# Mesenchymal Stem Cell-Based COVID-19 Therapy: Bioengineering Perspectives

**DOI:** 10.3390/cells11030465

**Published:** 2022-01-29

**Authors:** Nihal Karakaş, Süleyman Üçüncüoğlu, Damla Uludağ, Birnur Sinem Karaoğlan, Khalid Shah, Gürkan Öztürk

**Affiliations:** 1Department of Medical Biology, School of Medicine, İstanbul Medipol University, İstanbul 34810, Turkey; 2Regenerative and Restorative Medicine Research Center (REMER), Institute for Health Sciences and Technologies (SABITA), İstanbul Medipol University, İstanbul 34810, Turkey; sucuncuoglu@medipol.edu.tr (S.Ü.); duludag@st.medipol.edu.tr (D.U.); birnur.karaoglan@std.medipol.edu.tr (B.S.K.); gozturk@medipol.edu.tr (G.Ö.); 3Department of Biophysics, International School of Medicine, İstanbul Medipol University, İstanbul 34810, Turkey; 4Graduate School for Health Sciences, Medical Biology and Genetics Program, İstanbul Medipol University, İstanbul 34810, Turkey; 5Center for Stem Cell and Translational Immunotherapies, Brigham and Women’s Hospital, Harvard Medical School, Boston, MA 02114, USA; kshah@bwh.harvard.edu; 6Department of Physiology, International School of Medicine, İstanbul Medipol University, İstanbul 34810, Turkey

**Keywords:** mesenchymal stem cells, genetic engineering, bioengineering, COVID-19, SARS-CoV-2, ACE2, TMPRSS2

## Abstract

The novel pathogenic severe acute respiratory syndrome coronavirus 2 (SARS-CoV-2) causes coronavirus disease 2019 (COVID-19). Mesenchymal stem cells (MSCs) are currently utilized in clinics for pulmonary inflammatory diseases, including acute respiratory distress syndrome and acute lung injury. Given that MSCs offer a promising treatment against COVID-19, they are being used against COVID-19 in more than 70 clinical trials with promising findings. Genetically engineered MSCs offer promising therapeutic options in pulmonary diseases. However, their potential has not been explored yet. In this review, we provide perspectives on the functionally modified MSCs that can be developed and harnessed for COVID-19 therapy. Options to manage the SARS-CoV-2 infection and its variants using various bioengineering tools to increase the therapeutic efficacy of MSCs are highlighted.

## 1. Introduction

Severe acute respiratory syndrome coronavirus 2 (SARS-CoV-2) is an enveloped positive-sense RNA beta-coronavirus that has spread to almost all continents in just a few months [1]. The virus has been steadily rising worldwide since December 2019 and several variants have been discovered (https://www.who.int/en/activities/tracking-SARS-CoV-2-variants/, accessed on 22 November 2021). Despite the vaccines developed thus far against SARS-CoV-2, vaccine studies continue to be developed against variants that arise due to the rapid mutation of the virus [2,3,4,5]. Currently, there is no specific therapeutic intervention to control the dissemination of the disease.

To develop therapeutic interventions, the mechanism of the lethal infection was defined. SARS-CoV-2 primarily targets the ACE2 (angiotensin 1 converting enzyme 2)-expressing airway epithelial cells in the trachea, alveolar epithelial cells, vascular endothelial cells and pulmonary macrophages in the lung [6]. Other ACE2-expressing tissues, such as the heart, kidney, endothelium and particularly intestinal epithelial tissue, are also at risk of SARS-CoV-2 infection. These crucial organs are reported to be most affected by the virus, which reveals the need for comprehensive treatments [7,8].

To date, supplemental oxygen and mechanical ventilation support are mainly utilized as the standard care for COVID-19 in several countries. However, these approaches do not provide an effective way to diminish the systemic dysfunctional immune response [9,10]. These outcomes lead to COVID-19 mortality.

For an urgent and cost-effective solution to treat COVID-19, researchers have tried to repurpose potential drugs [11,12,13]. Although in vitro data of such potential treatment is promising, especially for mild cases, due to safety concerns randomized clinical trials are required [14,15,16].

In addition to evaluating potential drugs, vaccine studies against SARS-CoV-2 have been executed and functional benefits have been extensively reported [17,18,19,20,21,22]. The studies involve inactivated vaccines, live attenuated vaccines, nucleic acid vaccines, virus-like particles and viral vector vaccines. However, the rapid mutation of the virus is a big concern. To date, more than 30 mutations have been identified, which may negate candidate drugs, and these variations may cause lethal effects [23,24]. According to the World Health Organization (WHO), so far, the most rapidly transmitted and replicated variant is the B.1.617.2 (Delta). It has certain mutations in the spike (S) protein of the virus. Some important mutations in this variant, such as E484Q, L452R and P614R, facilitate the binding of S proteins to ACE2 receptors. For this reason, the development of variant-specific vaccines may be promising in terms of therapy management. Moreover, the existence of new mutations may alter the viral entry and function in an aggressive manner. This may then require the development of additional vaccine shots and potential new drugs. Considering these preventive/therapeutic limitations, COVID-19 may not be fully controlled [25,26]. Such reasons show the necessity of inclusive treatments against the forthcoming variants.

Considering all these limitations and the rapid increase in SARS-CoV-2 mutagenesis, efforts to establish alternative therapies are ultimately needed. Therefore, cell-based approaches can be utilized in parallel to conventional therapies. In particular, due to their immunosuppressive and anti-inflammatory features, mesenchymal stem cells (MSCs) are potential candidates to modulate the SARS-CoV-2-dependent cytokine storm. Moreover, MSCs were already in clinical trials for other pulmonary diseases such as acute respiratory distress syndrome (ARDS) and acute lung injury (ALI) [27,28,29,30]. Taken together, MSCs have been urged for clinical use in COVID-19 cases and many of the recent studies have reported their success in therapy [31,32,33,34,35].

What is more, to achieve an enhanced therapeutic efficacy, naive MSCs can be modulated. This strategy has been used for the treatment of several diseases, such as cancers, and a related approach is also being tested in clinical trials (NCT03298763, NCT02008539). To capacitate MSCs against COVID-19 and increase their effectiveness, here we suggest engineering those cells using gene-introducing/editing/regulatory systems and suggest possible technical tools. In this perspective, we also discuss the bioengineering tools and route of the modified MSCs against SARS-CoV-2 from bench to bedside.

## 2. Mesenchymal Stem Cell Therapy

Mesenchymal stem cells are multipotent and can specialize into several cell types from different lineages. Intravenously administered MSCs can migrate to sites of damaged tissue and promote angiogenesis, growth and differentiation of local progenitor cells [36,37,38,39,40]. They can also prevent apoptosis and microbial infection. MSCs can be utilized in clinics due to their regenerative, anti-inflammatory and immunomodulatory properties [41,42,43,44]. Some studies have shown that MSCs can be transplanted allogenically without a major risk of host immune response [45,46,47]. In addition, MSCs are easy to isolate and expand in culture from different tissues such as umbilical cord/cord blood, bone marrow and adipose tissue. These capabilities make MSCs a primary source for cell-based therapies in several diseases.

Preclinical and clinical studies indicate the efficacy of MSC therapy in myocardial infarction [48,49], diabetes [50], hepatic failure [51], acute graft versus host disease [52,53], pulmonary [54,55,56,57,58] and autoimmune diseases [59,60,61,62]. In particular, MSC therapy shows the reversal of symptoms in ARDS/ALI which supports the potential usage of MSCs against COVID-19. MSCs increase the secretion of angiopoietin-1, keratinocyte growth factor (FGF-7) [63], fibroblast growth factor-2 (FGF-2), vascular endothelial growth factor (VEGF), hepatocyte growth factor (HGF) [64,65] and a number of unique proteins and signaling molecules [66] which are pivotal in the restoration of tissue disrupted by COVID-19.

Recently, China, the United States, Spain, Iran and other countries began clinical trials using MSC therapy in COVID-19 cases. To date, 74 clinical trials are registered in the NIH clinical trial database (www.clinicaltrials.gov (accessed on 14 May 2021)), and there are also 15 registered clinical trials using the administration of MSCs to treat severe COVID-19 patients at http://www.chictr.org.cn (accessed on 14 May 2021).

The majority of the concluded clinical trials support MSC therapy against SARS-CoV-2 infection. No adverse effects on patients have been shown so far related to MSC transplantation [31,32,33,34,35,56,67]. Phase studies demonstrated prominent clinical outcomes in COVID-19 patients following MSC infusion, in terms of radiographic findings, inflammatory cytokine levels, liver function tests and pulmonary function indicators. In addition, a recent meta-analysis reported that MSC therapy holds substantial promise for the treatment of COVID-19 patients [68].

Although the clinical data accumulated is promising, there may be other factors to take into consideration. For instance, several studies have been conducted on the ACE2 expression levels and infection risk of MSCs by SARS-CoV-2 for therapeutic approaches. However, reported data is variable. In such a study, MSCs were reported to lack ACE2 and TMPRRS2 expressions [69,70,71], while other studies showed that ACE2 was highly expressed in MSCs derived from adult bone marrow, adipose tissue or umbilical cord [72,73]. Some studies also showed that ACE2 expression provides protective roles to MSCs against SARS-CoV-2 infection [74,75]. One of the concerns may be the sample numbers of each MSC source studied to make an overall conclusion. Although most of the researchers accept MSCs with or without expression of ACE2, extensive analysis of MSCs from a broad range of donors can provide a better insight into ACE2 expression patterns in MSCs. For instance, human MSCs derived from different sources (such as adipose, bone marrow, umbilical cord, cord blood, etc.), ethnical origins and individual differences (genetic or epigenetic background) can also show variations in the expression. These concerns should be clarified and MSC therapy may then be managed and improved accordingly. One solution to that may be the sorting of ACE2 negative populations in MSCs. Afterwards, these enhanced MSCs can be used for administration. This may protect MSCs from viral attack and prolong the survival of MSCs in the affected organ and therefore increase the MSC therapeutic efficacy against SARS-CoV-2.

## 3. Bioengineering Solutions to Improve MSC Therapy

MSC engineering strategies have been widely developed and applied to treat several pathologies, such as cancers [76,77,78,79,80,81]. The clinical potency and application of MSCs can be accelerated and broadened by various strategies such as small molecule priming, particle engineering and genetic engineering [82]. Immunomodulatory and anti-inflammatory factors may also be potentiated to extend the immunosuppressive profile of MSCs. Various genetic engineering strategies can be pursued to increase the therapeutic potential of MSCs (Figure 1).

The genetic engineering of MSCs can be designed to promote multiple abilities within the same MSC population. This provides a technical advantage to establish multimodal MSCs for therapeutic purposes against a certain disease [83,84,85]. Consequently, one possible way to enhance the therapeutic efficacy of MSCs can be to control MSC immunomodulatory/anti-inflammatory effects by managing cytokine production [82,86,87,88,89].

One of the most important clinical outcomes of SARS-CoV-2 infection is the development of a severe immune response with the onset of a cytokine storm. This results in a severe clinical picture and decreased patient survival. The disease is exacerbated by the excessive cytokine storm, especially in the intense infection environment of the lung tissue, affecting tissue homeostasis and the inability to provide intra-tissue immunomodulation. In this case, MSC therapy has a considerable potential in terms of its immunomodulatory properties. In particular, cytokine release can be balanced by immunomodulation in the damaged tissue which may allow the reduction of inflammatory responses. Our proposal aims to further strengthen the existing natural characteristics of MSCs with the aid of genetic engineering and to cope with this infection at a significant level. The cytokine targets that are active in hyperinflammation are IL-6, IL-8, TNF-α, IL1-β, MCP-1, GCS-F, IP-10, CCL1-3, IL-17 and IFN-γ [90,91,92,93]. In preclinical studies with MSC therapy for lung injury and ARDS, MSCs were reported to cause a decrease in pro-inflammatory cytokines, such as IL-6, IL-1α, IL-1β and IFN-γ, and an increase in anti-inflammatory cytokines, such as IL-4, IL-5 and IL-10 [29,94,95]. Taken together, the use of MSCs by amplifying anti-inflammatory and/or gene silencing/knockout of pro-inflammatory cytokines may be a potential approach against COVID-19 cases. One of the potential targets could be interleukin-6 since it is known as a major contributor in the cytokine storm in severe COVID-19 cases. Finally, modulating the cytokine release using bioengineering tools may increase therapeutic efficacy, which may ultimately improve the current clinical benefits of MSC administration.

Another bioengineering solution may be to target the SARS-CoV-2 infection mechanisms in MSCs. To date, the ACE2 expression analysis studies in MSCs were performed using in vitro monolayer culture systems and isolated viruses. Instead, in an affected organ, there might be different secretomes/mediators upon SARS-CoV-2 infection, and this may also lead to alterations of the viral entry/replication abilities. Accordingly, some researchers investigate if there are possible alternative routes for SARS-CoV-2 to infect the host cells. Recently, one study showed that in the human proximal tubular kidney cell line HK-2, there is an ACE2-independent way of SARS-CoV-2 infection in the host cells. Briefly, according to their results, soluble ACE2 (sACE2) (proteolytically cleaved by the infected cells and released into the affected organ) can make viral attacks possible in the cells even if they lack ACE2 expressions. Considering the presence of possible ACE2-independent infection mechanisms in a real environment, MSCs may be at risk of SARS-CoV-2 infection. Besides, new variants can develop alternative infection ways (gaining new random mutations may contribute to this). If this is the case, it may bring new discussions in. In which case, the proposed bioengineering strategies for MSCs can be expanded. For instance, the interaction with the host cell can be targeted. Engineering tools can help to protect us from the viral attacks by SARS-CoV-2 and its future more lethal variants.

Furthermore, considering the possibility of forthcoming more aggressive variants and alterations in the infection components (previously discussed in this review), MSCs can be potentiated to function through their protective roles using bioengineering tools with several different aspects [96,97,98,99,100].

### 3.1. Tools for MSC Engineering

There are two aspects of MSC engineering: the determination of the gene of interest for the desired MSC modification and the proper engineering technique. To engineer MSCs, direct delivery of plasmid DNA, pDNA (transfection) or viral delivery (transduction) may be preferred according to the experimental purposes. pDNA insertion in MSCs has been achieved by various techniques, such as nucleofection, electroporation or using commercially available transfection reagents. Critically, one of the concerns is that both transduction and transfection efficiencies may be variable. This can be modulated by introducing selection markers in plasmid constructs (such as resistance genes and fluorescence reporters). Mainly fluorescence-activated cell sorting or antibiotic treatment-based selection procedures can be followed to separate the proposed cell population.

MSC engineering, like that of any other human cell type, mainly requires a double stranded break (DSB) on the DNA. Initially, the DSBs on stem cells were achieved using site-directed zinc finger nucleases (ZFNs) [101] and transcription activator-like effector nucleases (TALENs) techniques [102]. However, the protein engineering requirements for the DNA binding domains of TALEN or ZFS systems was the biggest obstacle for widespread stem cell engineering [103]. In contrast, the CRISPR/Cas9 system offers a much simpler DNA binding mechanism by utilizing the guide RNA which complements the target DNA sequence [104]. It is also possible to target more than one gene by introducing more than one guide RNA during the CRISPR/Cas9 delivery. In addition to that, CRISPR/Cas9 systems can also be used to regulate the transcription by recruiting cleavage-free dead Cas9 (dCas9) enzymes coupled with the transcription factors [105].

Engineered MSCs can also be armed with suicide genes [106,107] such as HSV-TK/GCV (herpes simplex virus thymidine kinase/ganciclovir) and CD/5-FC (cytosine deaminase/5-Fluourocytosine; bacterial/yeast origin) to control unpredicted fate as the suicide genes have already been proposed in several clinical trials against cancers (NCT01913106 and NCT01172964) [108,109]. Principally, these genes code an enzyme converting the prodrug to its toxic metabolite. Transduced cells and neighboring untransduced cells, via the bystander effect, are the targets of the suicide gene/prodrug system. To limit the toxic effects of the system on other organs during construction, these suicide genes can be placed under an inducible or tissue-specific promoter. Alternatively, the inducible caspase-9-mediated suicide gene system creates a safety switch for engineered cell therapies. This strategy consists of a modified human caspase-9 fused to a human FK506-binding protein (FKBP). The chimeric protein has a high affinity to pharmaceutical small molecules (AP20187). Administration of the bioinert small molecule results in conditional dimerization and induces apoptosis in transduced cells [110,111].

#### 3.1.1. In Vitro Modeling

For in vitro COVID-19 modeling, a VERO E6 cell line (derived from African green monkey kidney) can be used to represent the infection environment since it is widely used in SARS-CoV-2 infection studies including vaccine developments. In addition to the VERO E6 cell, a study showed that, among a panel of cell lines tested, the HK-2 cell line can also be infected by SARS-CoV-2. Compared to VERO E6 cells, moderate and strong cytopathic effects have been reported in VERO E6 and HK-2 cells, respectively [112]. HK-2 cells may be a potential cell source for COVID-19 modeling since they are of human origin. Furthermore, the co-culture of engineered MSCs within a COVID-19 model may provide a reasonable assessment of therapeutic efficacy [113]. Viability of infected cells can be determined in a time-dependent manner. To distinguish viability of target cells from therapeutic MSCs, VERO E6 or HK-2 (a COVID-19 model) can be modified to express luciferases before transduction with SARS-CoV-2. Therefore, in the presence of substrate, a bioluminescence signal can be recorded selectively from luciferase-expressing cells. The infected cells can be detected by Western blotting for viral proteins or RT-PCR of viral mRNAs. In addition, SARS-CoV-2 infection is known for its overactivation of the innate immune system, and safety options of engineered MSC therapy should be considered. For this reason, upon the genetic engineering of enhanced immunomodulatory properties of MSCs, the cells are better to be examined for the increasing risk of a hyperinflammatory environment. To profile the changes, complete cytokine release in the conditioned media of MSCs can be measured by the ELISA method or using cytokine protein arrays (multiplex assays). Importantly, MSCs with increased anti-inflammatory properties should be optimized accordingly to balance the effects on natural immunity. Furthermore, considering the possibility of additional pro-inflammatory cytokine secretion, MSC-based cytokine release should be managed. Accordingly, the concentration of pro- and anti-inflammatory cytokines secreted by MSCs can be controlled by using inducible systems as previously mentioned. Next, these engineered cells can be administrated into certain animal models with active immune systems (such as syngeneic models) to evaluate the overactivation/inactivation of the immune system by modified MSCs only. This can then lead to the assessment of the therapeutic efficacy of modified MSCs in COVID-19 animal models. Consequently, the engineering strategy may increase the therapeutic efficacy of MSCs which are already in clinical trials for COVID-19 patients. This can be achieved by maintaining a pure MSC secretome and exhibiting variations that favor therapeutic effects. This may then provide a rationale for the in vivo evaluation of the therapy (Figure 2).

#### 3.1.2. Assessing Efficacy in COVID-19 Animal Models

Developing COVID-19 animal models for therapeutic and preventive approaches is also an ongoing area of current research since SARS-CoV-2 is a recent virus which needs to be explored. Several animal models mimicking COVID-19 pathology have been reported, and some are still in progress. Mice, Syrian hamsters, ferrets and non-human primates have been repurposed for COVID-19 animal models. In addition to these model animals, mink, cats, dogs, pigs, chicken, ducks and fruit bats are evaluated according to their potential aspects [114]. The lack of proper receptors in mouse models stands as the main obstacle for understanding the progression of SARS-CoV-2 and examining its response to various therapeutics. It was reported that SARS-CoV-2 cannot interact with the mouse ACE2 receptor effectively [115,116]. Various strategies have been developed to overcome this problem. These strategies include modifying the ACE2 receptor in mice to achieve binding to the SARS-CoV-2 S protein, and using humanized mouse models [117,118]. These efforts to develop a representative animal model are very important and necessary to elucidate the pathogenesis of the disease.

Upon choosing a suitable pathophysiological model for COVID-19, tracking the cells is essential to assess the fate of engineered MSCs, and this can be evaluated in multiple ways. For instance, bimodal vectors bearing fluorescent and bioluminescence agents enable researchers to track therapeutically engineered cells in several disease conditions such as cancers [119,120], bone defects [121], neurodegenerative disorders [122,123] and autoimmune diseases [124,125]. Accordingly, MSCs can be armed with labeled proteins and their survival in the pathologic microenvironment can be experimentally monitored [126,127]. Therefore, various bioengineering options can be applied to enhance the therapeutic efficacy of the MSC delivery. Thereupon, clinical settings can be optimized thoroughly. Alternatively, PET imaging may also be considered to track MSC fate in vivo after injection [128,129,130].

To measure the efficacy during the course of in vivo study, blood samples of models can be examined for white blood cell, neutrophil, T cell and lymphocyte counts. In addition, chest computed tomography (CCT) scans can be recruited to follow up the pneumonia relief. After the sacrifice and necropsy, blood and major organs can be analyzed for viral load and cytokine/chemokine profile. Furthermore, the live SARS-CoV-2 load can be measured by a median tissue culture infectious dose (TCID50) assay performed on the lung tissues [131]. After completing the requirements for in vivo studies, the effectiveness can be ultimately utilized for clinical translation.

### 3.2. Clinical Route of Engineered MSCs

MSCs are already in use for the treatment of various injuries of lower respiratory tracts, such as ARDS and ALI [28,132,133,134,135]. Recently, this system has been harnessed for COVID-19 cases [31,136,137].

According to the clinical trial database (clinicaltrials.gov), the MSC administration dose extends from 8 × 10^6^/dose up to 3 × 10^8^/dose and the median dose is 1 × 10^8^ cells/patient in the IV route. For clinical research, up to three doses of intravenous injection have been implemented in different promising trials [ChiCTR2000031139, ChiCTR2000030484, NCT04366063, NCT04390152, NCT04269525, NCT04400032]. A completed phase 1 clinical trial indicated that multiple intravenous infusions of MSCs do not adversely affect the course of COVID-19 disease and contributed to the decrease in all inflammatory cytokines within 14 days (NCT04252118). In addition, an open label dose escalation phase 1 trial demonstrated that infusion of 400 million umbilical cord-derived MSCs showed no dose-limiting toxicity [138].

All active clinical trials registered on clinicaltrials.gov use intravenous infusion to administer MSCs. In almost all cases, intravenously administered MSCs first accumulate in the lung vasculature, then move to other major organs, such as the liver and kidneys, and after a variable but short period (24 h to 14 days) are no longer detectable in the body [139]. A phase 1 trial including 11 COVID-19 patients showed that there is a significant reduction in serum TNF-α, CRP and IFN-γ levels following MSC infusion. Moreover, radiological findings of CT scans support the clinical improvements upon MSC administration. The local and systemic clinical outcomes support the idea that IV infusion of MSCs alleviates inflammation and contributes to the recovery process. Another phase 2 trial conducted on 101 patients showed a significant decline in lung lesion volume in COVID-19 patients treated with MSCs administration compared with the placebo group (NCT04288102). According to the same phase study, the restoration of lung function was significantly better in the MSC-treated group.

Besides MSC infusion, there are 7 active clinical trials registered on clinicaltrials.gov on the administration of MSC secretome and extracellular vesicles in the treatment of COVID-19 (NCT04491240, NCT04602442, NCT04276987, NCT04798716, NCT04657458, NCT04753476, NCT04384445). Secretome is defined as the set of bioactive factors, such as enzymes, soluble regulatory proteins and growth factors, secreted by a specific type of cell [140]. The main secretome preparation can be performed either via concentration of a conditioned medium or by collecting extracellular vesicles [141]. An MSC-derived conditioned medium consists of both the exosomes, microvesicles and the soluble factors of secretome. A conditioned medium can be fractioned via centrifugation, filtration, polymer precipitation-based methodologies and chromatography to differentiate vesicles from soluble components. Secretome injection showed similar therapeutic effects to MSCs administration. Furthermore, they can also bypass some of the side effects of intact cell infusion [142]. In line with this, a phase 1 trial aiming to study the therapeutic effect of MSC-derived exosome inhalation in SARS-CoV-2 infections showed no adverse effects related with exosome administration (NCT04491240). Another phase study indicated improved laboratory findings and decreased acute phase reactants in COVID-19 patients upon bone marrow MSC-derived exosome inhalation [143].

In addition to the therapeutic effects MSCs offer, the possible risks of MSC transplantation should be considered to create a more reliable and complete view. In general, the potential risks of the clinical administration of MSCs comprise a pro-coagulant profile of MSCs [144,145], unexpected pro-inflammatory effects, and disturbed differentiation and pro-neoplastic capacities after injection [146].

MSC-based products express variable levels of highly procoagulant tissue factor (TF/CD142), creating a safety concern for the IV infusion of MSCs [147]. To prevent thromboembolism in at-risk patients, anticoagulants can be initiated as adjunct therapy. Alternative routes of cell administration can be exploited, such as intramuscular injection leading the cells into the extravascular space directly.

Taken together, proposing genetically modified MSCs may provide an improved MSC-based therapy for COVID-19. A potential concern in clinical settings over reducing the pro-inflammatory properties and/or enhancing the anti-inflammatory properties of MSCs may be the excessive suppression of the innate immune system. In this case, the immunomodulating effects of MSCs can be achieved by constructing gene regulatory systems. For instance, inducible systems can be used to manage the initiation and termination of the signal necessary to control the immunomodulation.

Even though it is hard to predict how the engineered MSCs will respond to all these regulatory dynamics, it is much more about hope rather than hype when considering current attempts of engineered MSC therapy in several diseases.

## 4. Conclusions

The recent preclinical data and phase trials suggest MSCs as potential therapy vehicles for COVID-19 cases. Following a brief discussion on possible engineering scenarios, here we suggest a prospective solution for the continuous and controllable secretion of anti-inflammatory cytokines by modified MSCs which may then enhance the therapy against SARS-CoV-2. This moldable approach mainly targets SARS-CoV-2-based deadly infection. The strategy can also be versatile in managing altered cytokine storms caused by the new probable variants. Further studies will provide a better understanding to assess the efficiency of our proposed hypothesis.

## Figures and Tables

**Figure 1 cells-11-00465-f001:**
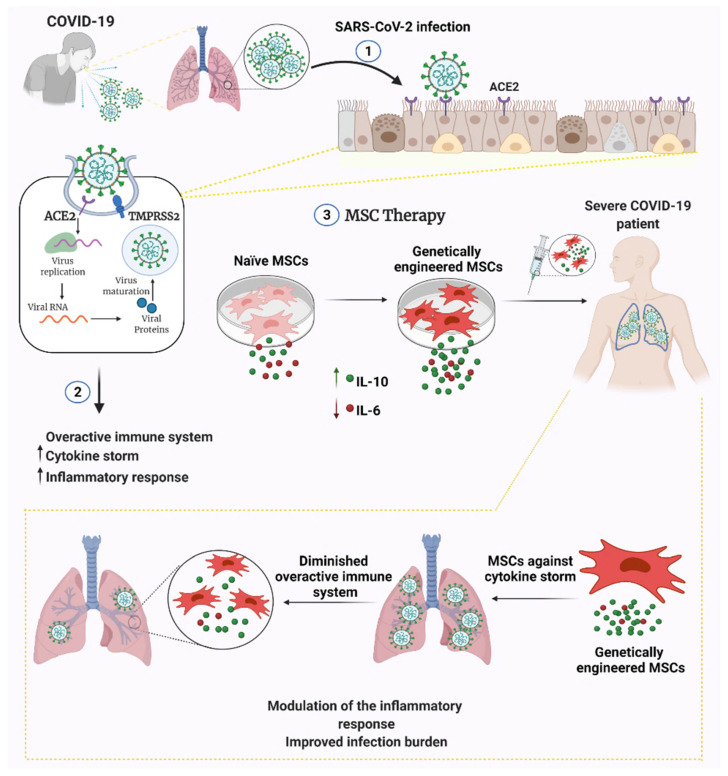
Engineered versus naive MSC therapy against COVID-19. MSCs can be engineered to secrete anti-inflammatory cytokines and/or genetically silenced for pro-inflammatory cytokine release. Engineering MSCs may be capacitated to prevent infection and prolong MSC longevity/functionality within the patient. Abbreviations: ACE2, angiotensin-converting enzyme 2; TMPRSS2, transmembrane serine protease 2; IL-10, interleukin-10; IL-6, interleukin-6.

**Figure 2 cells-11-00465-f002:**
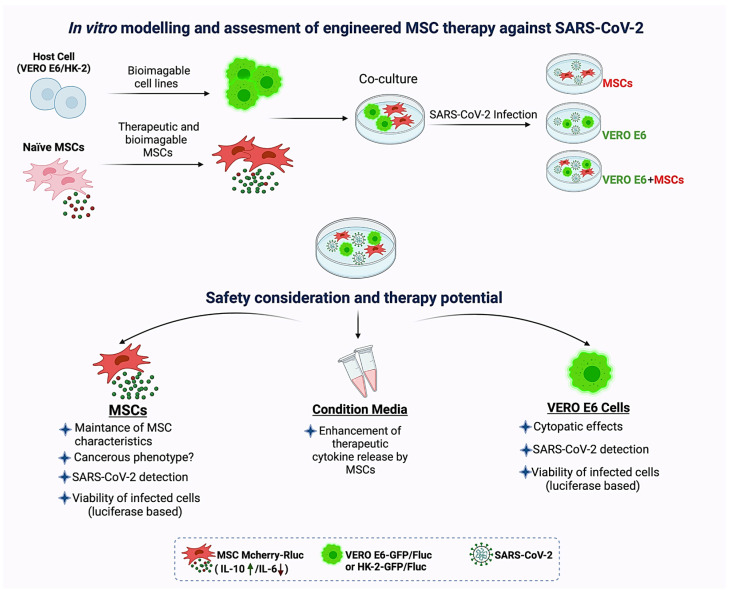
In vitro modeling and assessment of engineered MSC therapy against SARS-CoV-2. Co-culture of therapeutically engineered MSCs and bioimageable host cells (VERO E6 or HK-2 cells). Groups: includes MSC only, host cells only (VERO E6/HK-2) and their co-cultures (MSC+VERO E6/HK-2). All of these groups can be infected by SARS-CoV-2, while the control groups remain unaffected. MSCs in co-cultures with exposure to SARS-CoV-2 can be examined for their safety and therapeutic potential. For this, MSCs in co-culture can be evaluated for their (1) MSC-specific characteristics (expression of surface markers, capacity of mesodermal differentiation and plasticity abilities), (2) risks of cancerogenic phenotype (such as expressing cancer stem cell markers, forming tumorspheres in vitro, and teratomes in vivo, and vice versa) and (3) their survival rates in the infectious microenvironment. Additionally, the conditioned medium can be assayed for enhanced anti-inflammatory/immunomodulatory cytokine release (or any other proposed secretions) by the engineered MSCs. To confirm the infection, SARS-CoV-2 and relative cytopathic effects (CPE) in host cells (VERO E6/HK-2) can be detected. The viability can be measured in host cells which have been previously modified to express one of the luciferases. Likewise, fluorescence proteins can be integrated to the luciferase (during pDNA construction) and the cells in co-cultures can be distinguishable from MSCs under the microscope. Conversely, bioimageable MSCs (or both MSCs and host cells) can be established according to the experimental necessities. Options can be altered and such an infectious coculture system can mimic MSC residing in the infectious microenvironment and their interactions with the hosts. This may eventually provide an in vitro model for various evaluations.

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
