# Peer review of "Mesenchymal Stem Cell-Based COVID-19 Therapy: Bioengineering Perspectives"

_cells, 2022, doi:10.3390/cells11030465_

Round 1

Reviewer 1 Report

Öztürk et al proposed an innovative MSCs-based approach aimed to COVID-19 therapy. The idea is original and challenging, however it lacks evidences and the study design is not clear.

In line 60 authors say" In addition, cases of infection have been reported in vaccinated people, with or without SARS-CoV-2 infection. I don't understand the meaning of the sentence. In line 61 is reported "The Covid-19 pandemic continues due to the exposure of vaccinated individuals to infection and developing variants [17–20]". In my opinion, the citations are not related to the concept that the authors want to express, and I don't think that vaccinated subjects contribute to the progression of the pandemic.

As in vitro model, authors proposed VERO6 or HAECs cell lines, but, I think, the proposals are not equivalent. It is known that SARS-Cov2 is able to infect non-human primate cells leading to a marked damage as cytopathic effects, evidence not seen in other cell lines.  DOI:https://doi.org/10.1016/S2666-5247(20)30004-5

To measured viability of infected cells they suggest to modify cell model to express luciferase, while measure the therapeutic efficacy qPCR on media culture. How did the authors supposed to modify cells? By transfection? And how MSCs in co-culture can modulate the immune response if the immune cells are not present?

The work is not clear, the experimental design is not well illustrated, some steps are missing.

Reviewer 2 Report

MSC therapy offers promising opportunities for COVID-19 patients. To enhance the therapeutic potential and improve the outcomes after transplantation, genetically engineered MSC therapy has been applied in recent years to treat several human diseases. This perspective mainly focuses on approaches to COVID-19 therapy using genetically engineered MSCs. This is an interesting paper on an important topic. However, I have some suggestions and comments which may improve the quality of this paper.

P1, line 21: Please do not indicate abbreviations (ARDS) and (ALI) in the abstract, because further in the abstract they are not used

P1, line 35: Pay attention to the spaces here: “existed(https://www.who.int/en/activities/tracking-SARS-CoV-2-35 variants/)”

P1, line 44: Use “,” instead of “;”

P2, lines 55-56: Should it be: “In addition to evaluating potential drugs, vaccine studies continue to be developed against SARS-CoV-2.” instead of “In addition to evaluating potential drugs, vaccine studies continue to be developed. Against SARS-CoV-2;”?

P2, line 60: Please, clarify this statement: “In addition, cases of infection have been reported in vaccinated persons, with or without SARS-CoV-2 infection”

P2, line 61: And this statement also need to be clarify: “The Covid-19 pandemic continues due to the exposure of vaccinated individuals to infection and developing variants”

P2, line 69: Use “ACE2” instead of “ACE-2” as earlier in the text

P3, lines 104: Abbreviations “VEGF, HGF” should be defined

P3, line 113: Pay attention to the spaces here: “plantation[29–33,54,65].”

P4, line 126: Unnecessary period

P4, Figure 1: Use “COVID19” instead of “Covid19”, “SARS-CoV-2” instead of “SARS-Cov2” as earlier in the text. An abbreviation “TMPRSS2” should be defined

P4, line 148: Use “,” instead of “;”

P4, lines 148-149: Perhaps "IL-6" should appear first in this list of "cytokines such as IL-1α, IL-1β, IL-6, IFN-γ" as the authors discuss its important role later in the text.

P5, line 163: Use “,” instead of “;”

P5, line 191: Use “,” instead of “;”

P5-6, lines 200-210: The authors discuss the post-injection tracking of MSCs and mentioned “various bioengineering options” that “can be applied to enhance therapeutic efficacy of the MSC delivery.” It should be clarified by what methods. Do they include the introduction of suicidal genes described later in the text (pp. 260-273)?

P6, line 215: Please check this out: “infectious(edit)”

P6, line 216: An abbreviation “TCID50” should be defined

P6, line 217: Use “ALI” instead of “acute lung injuries”

P6, line 224: Please, check out if this correct: “the MSC administration dose extends from 8 x 10^6/dose up to 5x10^6/kg.”

P6, line 229: Pay attention to the space and period here: “14 days(NCT04252118)”

P6, line 231: The unnecessary abbreviation “(UC-MSCs)” is not used further in the text.

P6, line 248: Missing period: “NCT04384445) In a phase”

P7, line 264: Please move references to the end, after “(NCT01913106 and NCT01172964)”

P7, line 278: Please, check out if “make as” in this statement correct: “The recent preclinical data and phase trials make MSCs as potential therapy vehicles  for COVID-19 cases.”

Reviewer 3 Report

Comments to the authors (Cells-150054):

In this review manuscript the authors discussed the potential protective function of bioengineered MSCs against SARS-Cov-2 infection. The topic is very interesting in the field of COVID-19 therapy. However, there are couple of questions related to this study; if included, would further enhance the manuscript.

  • In introduction section, lines 75-77, authors talked about the lack of expression of ACE2 in MSCs, however there are studies out there showing that MSCs exert their protective role through ACE2/MasR axis of RAS (e.g., Barzegar et al. Stem Cell (2021)). They may need to justify this more clearly or discuss the different sources from which MSCs can be isolated.

  • In mesenchymal stem cell therapy section, the authors only focused on the beneficial effects of intravenously injected MSCs, however, the major challenge regarding IV administration of stem cells particularly in clinical trials is the high risk of coagulation since MSCs or MSCs-derived extracellular vesicles could trigger the coagulation pathway by recruitment and activation of tissue factor (TF) on the surface of MSCs. Moreover, it has been shown that MSCs could be entrapped in the lung or accumulated in the spleen after IV injection which would eliminate the efficacy of stem cell therapy. The authors need to extensively discuss these challenges in the manuscript.

  • In line 157, please introduce VERO E6 and HAECs cells more clearly and explain the rational for the use of these cells as an in vitro COVID-19 model (e.g., highly susceptible to virus infection, easy to grow, manipulate, …).

  • In line 170-171, it would be more acceptable if the authors could name the techniques that can be used to measure the MSC-released cytokine profile.

  • Additionally, I would like to suggest the authors to briefly introduce the conditioned media, secretome and extracellular vesicles.

  • More importantly, the authors need to include another section to introduce, discuss, and compare the available approaches are currently being used to engineer or manipulate the MSCs (e.g., gene delivery, transduction, genome editing like CRISPER and TALEN)!!!

  • I really liked the discussion about COVID-19 animal models, and the way the authors explained the strategies to overcome the limitations of mouse model!

  • In general, the manuscript needs an extensive English, grammar and punctuation editing to merit publication in Cells.

Round 2

Reviewer 1 Report

The authors improved the manuscript and clarified all carent points.

Reviewer 3 Report

Thanks to authors for addressing all the comments point by point. I believe the manuscript has been sufficiently improved to be published in "Cells".